# Variational Graph Recurrent Neural Networks

**Ehsan Hajiramezanali**[†*]**, Arman Hasanzadeh**[†*]**, Nick Duffield**[†]**, Krishna Narayanan**[†]**,
Mingyuan Zhou**[‡]**, Xiaoning Qian**[†]

† Department of Electrical and Computer Engineering, Texas A&M University
{ehsanr, armanihm, duffieldng, krn, xqian}@tamu.edu
‡ McCombs School of Business, The University of Texas at Austin
mingyuan.zhou@mccombs.utexas.edu

## Abstract

Representation learning over graph structured data has been mostly studied in static graph settings while efforts for modeling dynamic graphs are still scant. In this paper, we develop a novel hierarchical variational model that introduces additional latent random variables to jointly model the hidden states of a graph recurrent neural network (GRNN) to capture both topology and node attribute changes in dynamic graphs. We argue that the use of high-level latent random variables in this variational GRNN (VGRNN) can better capture potential variability observed in dynamic graphs as well as the uncertainty of node latent representation. With semi-implicit variational inference developed for this new VGRNN architecture (SI-VGRNN), we show that flexible non-Gaussian latent representations can further help dynamic graph analytic tasks. Our experiments with multiple real-world dynamic graph datasets demonstrate that SI-VGRNN and VGRNN consistently outperform the existing baseline and state-of-the-art methods by a significant margin in dynamic link prediction.

## 1 Introduction

Node embedding maps each node in a graph to a vector in a low-dimensional latent space, in which classical feature vector-based machine learning formulations can be adopted [5]. Most of the existing node embedding techniques assume that the graph is static and that learning tasks are performed on fixed sets of nodes and edges [19, 23, 12, 20, 14, 1]. However, many real-world problems are modeled by *dynamic* graphs, where graphs are constantly evolving over time. Such graphs have been typically observed in social networks, citation networks, and financial transaction networks. A naive solution to node embedding for dynamic graphs is simply applying static methods to each snapshot of dynamic graphs. Among many potential problems of such a naive solution, it is clear that it ignores the temporal dependencies between snapshots.

Several node embedding methods have been proposed to capture the temporal graph evolution for both networks without attributes [10, 26] and attributed networks [24, 16]. However, all of the existing dynamic graph embedding approaches represent each node by a deterministic vector in a low-dimensional space [2]. Such deterministic representations lack the capability of modeling uncertainty of node embedding, which is a natural consideration when having multiple information sources, i.e. node attributes and graph structure.

In this paper, we propose a novel node embedding method for dynamic graphs that maps each node to a random vector in the latent space. More specifically, we first introduce a dynamic graph autoencoder model, namely graph recurrent neural network (GRNN), by extending the use of graph convolutional

---

neural networks (GCRN) [21] to dynamic graphs. Then, we argue that GRNN lacks the expressive power for fully capturing the complex dependencies between topological evolution and time-varying node attributes because the output probability in standard RNNs is limited to either a simple unimodal distribution or a mixture of unimodal distributions [3, 22, 6, 8]. Next, to increase the expressive power of GRNN in addition to modeling the uncertainty of node latent representations, we propose variational graph recurrent neural network (VGRNN) by adopting high-level latent random variables in GRNN. Our proposed VGRNN is capable of learning interpretable latent representation as well as better modeling of very sparse dynamic graphs.

To further boost the expressive power and interpretability of our new VGRNN method, we integrate semi-implicit variational inference [25] with VGRNN. We show that semi-implicit variational graph recurrent neural network (SI-VGRNN) is capable of inferring more flexible and complex posteriors. Our experiments demonstrate the superior performance of VGRNN and SI-VGRNN in dynamic link prediction tasks in several real-world dynamic graph datasets compared to baseline and state-of-the-art methods.

## 2  Background

**Graph convolutional recurrent networks (GCRN).** GCRN was introduced by Seo et al. [21] to model time series data defined over nodes of a static graph. Series of frames in videos and spatio-temporal measurements on a network of sensors are two examples of such datasets. GCRN combines graph convolutional networks (GCN) [4] with recurrent neural networks (RNN) to capture spatial and temporal patterns in data. More precisely, given a graph $G$ with $N$ nodes, whose topology is determined by the adjacency matrix $\mathbf{A} \in \mathbb{R}^{N \times N}$, and a sequence of node attributes $\mathcal{X} = \{\mathbf{X}^{(1)}, \mathbf{X}^{(2)}, \ldots, \mathbf{X}^{(T)}\}$, GCRN reads $M$-dimensional node attributes $\mathbf{X}^{(t)} \in \mathbb{R}^{N \times M}$ and updates its hidden state $\mathbf{h}_t \in \mathbb{R}^p$ at each time step $t$:

$$\mathbf{h}_t = f\left(\mathbf{A}, \mathbf{X}^{(t)}, \mathbf{h}_{t-1}\right). \tag{1}$$

Here $f$ is a non-probabilistic deep neural network. It can be any recursive network including gated activation functions such as long short-term memory (LSTM) or gated recurrent units (GRU), where the deep layers inside them are replaced by graph convolutional layers. GCRN models node attribute sequences by parameterizing a factorization of the joint probability distribution as a product of conditional probabilities such that

$$p\left(\mathbf{X}^{(1)}, \mathbf{X}^{(2)}, \ldots, \mathbf{X}^{(T)} \mid \mathbf{A}\right) = \prod_{t=1}^{T} p\left(\mathbf{X}^{(t)} \mid \mathbf{X}^{(<t)}, \mathbf{A}\right); \quad p\left(\mathbf{X}^{(t)} \mid \mathbf{X}^{(<t)}, \mathbf{A}\right) = g(\mathbf{A}, \mathbf{h}_{t-1}).$$

Due to the deterministic nature of the transition function $f$, the choice of the mapping function $g$ here effectively defines the only source of variability in the joint probability distributions $p(\mathbf{X}^{(1)}, \mathbf{X}^{(2)}, \ldots, \mathbf{X}^{(T)} \mid \mathbf{A})$ that can be expressed by the standard GCRN. This can be problematic for sequences that are highly variable. More specifically, when the variability of $X$ is high, the model tries to map this variability in hidden states $h$, leading to potentially high variations in $h$ and thereafter overfitting of training data. Therefore, GCRN is not fully capable of modeling sequences with high variations. This fundamental problem of autoregressive models has been addressed for non-graph-structured datasets by introducing stochastic hidden states to the model [7, 3, 9].

In this paper, we integrate GCN and RNN into a graph RNN (GRNN) framework, which is a dynamic graph autoencoder model. While GCRN aims to model dynamic node attributes defined over a static graph, GRNN can get different adjacency matrices at different time snapshots and reconstruct the graph at time $t$ by adopting an inner-product decoder on the hidden state $\mathbf{h}_t$. More specifically, $\mathbf{h}_t$ can be viewed as node embedding of the dynamic graph at time $t$. To further improve the expressive power of GRNN, we introduce stochastic latent variables by combining GRNN with variational graph autoencoder (VGAE) [14]. This way, not only we can capture time dependencies between graphs without making smoothness assumption, but also each node is represented with a distribution in the latent space. Moreover, the prior construction devised in VGRNN allows it to predict links in the future time steps.

**Semi-implicit variational inference (SIVI).** SIVI has been shown effective to learn posterior distributions with skewness, kurtosis, multimodality, and other characteristics, which were not captured

by the existing variational inference methods [25]. To characterize the latent posterior $q(\mathbf{z}|\mathbf{x})$, SIVI introduces a mixing distribution on the parameters of the original posterior distribution to expand the variational family with a hierarchical construction: $\mathbf{z} \sim q(\mathbf{z}|\psi)$ with $\psi \sim q_\phi(\psi)$. $\phi$ denotes the distribution parameter to be inferred. While the original posterior $q(\mathbf{z}|\psi)$ is required to have an analytic form, its mixing distribution is not subject to such a constraint, and so the marginal posterior distribution is often implicit and more expressive that has no analytic density function. It is also common that the marginal of the hierarchy is implicit, even if both the posterior and its mixing distribution are explicit. We will integrate SIVI in our new model to infer more flexible and interpretable node embedding for dynamic graphs.

## 3    Variational graph recurrent neural network (VGRNN)

### 3.1    Overview

We consider a dynamic graph $\mathcal{G} = \{G^{(1)}, G^{(2)}, \ldots, G^{(T)}\}$ where $G^{(t)} = (\mathcal{V}^{(t)}, \mathcal{E}^{(t)})$ is the graph at time step $t$ with $\mathcal{V}^{(t)}$ and $\mathcal{E}^{(t)}$ being the corresponding node and edge sets, respectively. In this paper, we aim to develop a model that is universally compatible with potential changes in both node and edge sets. In particular, the cardinality of both $\mathcal{V}^{(t)}$ and $\mathcal{E}^{(t)}$ can change across time. There are no constraints on the relationships between $(\mathcal{V}^{(t)}, \mathcal{E}^{(t)})$ and $(\mathcal{V}^{(t+1)}, \mathcal{E}^{(t+1)})$, namely new nodes can join the dynamic graph and create edges to the existing nodes or previous nodes can disappear from the graph. On the other hand, new edges can form between snapshots while existing edges can disappear. Let $N_t$ denotes the number of nodes , i.e., the cardinality of $\mathcal{V}^{(t)}$, at time step $t$. Therefore, VGRNN can take as input a variable-length adjacency matrix sequence $\mathcal{A} = \{\mathbf{A}^{(1)}, \mathbf{A}^{(2)}, \ldots, \mathbf{A}^{(T)}\}$. In addition, when considering node attributes, different attributes can be observed at different snapshots with a variable-length node attribute sequence $\mathcal{X} = \{\mathbf{X}^{(1)}, \mathbf{X}^{(2)}, \ldots, \mathbf{X}^{(T)}\}$. Note that $\mathbf{A}^{(t)}$ and $\mathbf{X}^{(t)}$ are $N_t \times N_t$ and $N_t \times M$ matrices, respectively, where $M$ is the dimension of the node attributes that is constant across time. Inspired by variational recurrent neural networks (VRNN) [3], we construct VGRNN by integrating GRNN and VGAE so that complex dependencies between topological and node attribute dynamics are modeled sufficiently and simultaneously. Moreover, each node at each time is represented with a distribution, hence uncertainty of latent representations of nodes are also modelled in VGRNN.

### 3.2    VGRNN model

**Generation.**    The VGRNN model adopts a VGAE to model each graph snapshot. The VGAEs across time are conditioned on the state variable $\mathbf{h}_{t-1}$, modeled by a GRNN. Such an architecture design will help each VGAE to take into account the temporal structure of the dynamic graph. More critically, unlike a standard VGAE, our VGAE in VGRNN takes a new prior on the latent random variables by allowing distribution parameters to be modelled by either explicit or implicit complex functions of information of the previous time step. More specifically, instead of imposing a standard multivariate Gaussian distribution with deterministic parameters, VGAE in our VGRNN learns the prior distribution parameters based on the hidden states in previous time steps. Hence, our VGRNN allows more flexible latent representations with greater expressive power that captures dependencies between and within topological and node attribute evolution processes. In particular, we can write the construction of the prior distribution adopted in our experiments as follows,

$$p\left(\mathbf{Z}^{(t)}\right) = \prod_{i=1}^{N_t} p\left(\mathbf{Z}_i^{(t)}\right); \ \ \mathbf{Z}_i^{(t)} \sim \mathcal{N}\left(\boldsymbol{\mu}_{i,\text{prior}}^{(t)}, \text{diag}((\boldsymbol{\sigma}_{i,\text{prior}}^{(t)})^2)\right), \ \ \left\{\boldsymbol{\mu}_{\text{prior}}^{(t)}, \boldsymbol{\sigma}_{\text{prior}}^{(t)}\right\} = \varphi^{\text{prior}}(\mathbf{h}_{t-1}),$$
(2)

where $\boldsymbol{\mu}_{\text{prior}}^{(t)} \in \mathbb{R}^{N_t \times l}$ and $\boldsymbol{\sigma}_{\text{prior}}^{(t)} \in \mathbb{R}^{N_t \times l}$ denote the parameters of the conditional prior distribution, and $\boldsymbol{\mu}_{i,\text{prior}}^{(t)}$ and $\boldsymbol{\sigma}_{i,\text{prior}}^{(t)}$ are the $i$-th row of $\boldsymbol{\mu}_{\text{prior}}^{(t)}$ and $\boldsymbol{\sigma}_{\text{prior}}^{(t)}$, respectively. Moreover, the generating distribution will be conditioned on $\mathbf{Z}^{(t)}$ as:

$$\mathbf{A}^{(t)} \,|\, \mathbf{Z}^{(t)} \sim \text{Bernoulli}\left(\pi^{(t)}\right), \quad \pi^{(t)} = \varphi^{\text{dec}}\left(\mathbf{Z}^{(t)}\right),$$
(3)

where $\pi^{(t)}$ denotes the parameter of the generating distribution; $\varphi^{\text{prior}}$ and $\varphi^{\text{dec}}$ can be any highly flexible functions such as neural networks.

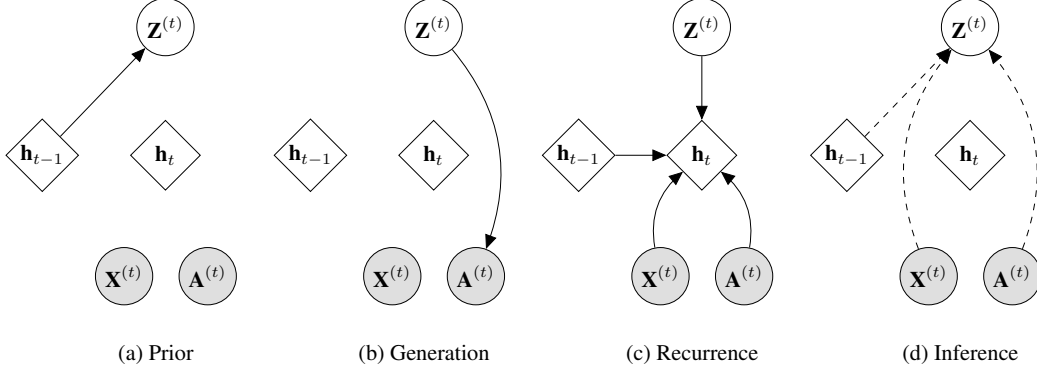

|  (a) Prior | (b) Generation | (c) Recurrence | (d) Inference |

Figure 1: Graphical illustrations of each operation of VGRNN; (a) computing the conditional prior by (2); (b) decoder function (3); (c) updating the GRNN hidden states using (4); and (d) inference of the posterior distribution for latent variables by (3.2).

On the other hand, the backbone GRNN enables flexible modeling of complex dependency involving both graph topological dynamics and node attribute dynamics. The GRNN updates its hidden states using the recurrence equation:

$$\mathbf{h}_t = f\left(\mathbf{A}^{(t)}, \varphi^{\mathbf{x}}\left(\mathbf{X}^{(t)}\right), \varphi^{\mathbf{z}}\left(\mathbf{Z}^{(t)}\right), \mathbf{h}_{t-1}\right), \tag{4}$$

where $f$ is originally the transition function from equation (1). Unlike the GRNN defined in [21], graph topology can change in different time steps as it does in real-world dynamic graphs, and the adjacency matrix $\mathbf{A}^{(t)}$ is time dependent in VGRNN. To further enhance the expressive power, $\varphi^{\mathbf{x}}$ and $\varphi^{\mathbf{z}}$ are deep neural networks which operate on each node independently and extract features from $\mathbf{X}^{(t)}$ and $\mathbf{Z}^{(t)}$, respectively. These feature extractors are crucial for learning complex graph dynamics. Based on (4), $\mathbf{h}_t$ is a function of $\mathbf{A}^{\leq(t)}$, $\mathbf{X}^{\leq(t)}$, and $\mathbf{Z}^{\leq(t)}$. Therefore, the prior and generating distributions in equations (2) and (3) define the distributions $p(\mathbf{Z}^{(t)} \mid \mathbf{A}^{(<t)}, \mathbf{X}^{(<t)}, \mathbf{Z}^{(<t)})$ and $p(\mathbf{A}^{(t)} \mid \mathbf{Z}^{(t)})$, respectively. The generative model can be factorized as

$$p\left(\mathbf{A}^{(\leq T)}, \mathbf{Z}^{(\leq T)} \mid \mathbf{X}^{(<T)}\right) = \prod_{t=1}^{T} p\left(\mathbf{Z}^{(t)} \mid \mathbf{A}^{(<t)}, \mathbf{X}^{(<t)}, \mathbf{Z}^{(<t)}\right) p\left(\mathbf{A}^{(t)} \mid \mathbf{Z}^{(t)}\right), \tag{5}$$

where the prior of the first snapshot is considered to be a standard multivariate Gaussian distribution, i.e. $p(\mathbf{Z}_i^{(0)} \mid -) \sim \mathcal{N}(0, \mathbf{I})$ for $i \in \{1, \ldots, N_0\}$ and $\mathbf{h}_0 = \mathbf{0}$. Also, if a previously unobserved node is added to the graph at snapshot $t$, we consider the hidden state of that node at snapshot $t-1$ is zero and hence the prior for that node at time $t$ is $\mathcal{N}(0, \mathbf{I})$. If node deletion occurs, we assume that the identity of nodes can be maintained thus removing a node, which is equivalent to removing all the edges connected to it, will not affect the prior construction for the next step. More specifically, the sizes of $\mathbf{A}$ and $\mathbf{X}$ can change in time while their latent space maintains across time.

**Inference.** With the VGRNN framework, the node embedding for dynamic graphs can be derived by inferring the posterior distribution of $\mathbf{Z}^{(t)}$ which is also a function of $\mathbf{h}_{t-1}$. More specifically,

$$q\left(\mathbf{Z}^{(t)} \mid \mathbf{A}^{(t)}, \mathbf{X}^{(t)}, \mathbf{h}_{t-1}\right) = \prod_{i=1}^{N_t} q\left(\mathbf{Z}_i^{(t)} \mid \mathbf{A}^{(t)}, \mathbf{X}^{(t)}, \mathbf{h}_{t-1}\right) = \prod_{i=1}^{N_t} \mathcal{N}\left(\boldsymbol{\mu}_{i,\text{enc}}^{(t)}, \text{diag}((\boldsymbol{\sigma}_{i,\text{enc}}^{(t)})^2)\right),$$

$$\boldsymbol{\mu}_{\text{enc}}^{(t)} = \text{GNN}_\mu\left(\mathbf{A}^{(t)}, \text{CONCAT}\left(\varphi^{\mathbf{x}}\left(\mathbf{X}^{(t)}\right), \mathbf{h}_{t-1}\right)\right),$$

$$\boldsymbol{\sigma}_{\text{enc}}^{(t)} = \text{GNN}_\sigma\left(\mathbf{A}^{(t)}, \text{CONCAT}\left(\varphi^{\mathbf{x}}\left(\mathbf{X}^{(t)}\right), \mathbf{h}_{t-1}\right)\right), \tag{6}$$

where $\boldsymbol{\mu}_{\text{enc}}^{(t)}$ and $\boldsymbol{\sigma}_{\text{enc}}^{(t)}$ denote the parameters of the approximated posterior, and $\boldsymbol{\mu}_{i,\text{enc}}^{(t)}$ and $\boldsymbol{\sigma}_{i,\text{enc}}^{(t)}$ are the $i$-th row of $\boldsymbol{\mu}_{\text{enc}}^{(t)}$ and $\boldsymbol{\sigma}_{\text{enc}}^{(t)}$, respectively. $\text{GNN}_\mu$ and $\text{GNN}_\sigma$ are the encoder functions and can be any of the various types of graph neural networks, such as GCN [15], GCN with Chebyshev filters [4] and GraphSAGE [13].

**Learning.** The objective function of VGRNN is derived from the variational lower bound at each snapshot. More precisely, using equation (5) , the evidence lower bound of VGRNN can be written as follows,

$$
\mathcal{L} = \sum_{t=1}^{T} \Bigg\{ \mathbb{E}_{\mathbf{Z}^{(t)} \sim q\left(\mathbf{Z}^{(t)} \mid \mathbf{A}^{(\leq t)}, \mathbf{X}^{(\leq t)}, \mathbf{Z}^{(<t)}\right)} \log p\left(\mathbf{A}^{(t)} \mid \mathbf{Z}^{(t)}\right)
$$
$$
- \mathbf{KL}\bigg( q\left(\mathbf{Z}^{(t)} \mid \mathbf{A}^{(\leq t)}, \mathbf{X}^{(\leq t)}, \mathbf{Z}^{(<t)}\right) \| p\left(\mathbf{Z}^{(t)} \mid \mathbf{A}^{(<t)}, \mathbf{X}^{(<t)}, \mathbf{Z}^{(<t)}\right) \bigg) \Bigg\}. \tag{7}
$$

We learn the parameters of the generative and inference models jointly by optimizing the variational lower bound with respect to the variational parameters. The graphical representation of VGRNN is illustrated in Fig. 1, operations (a)–(d) correspond to equations (2) – (4), and (3.2), respectively. We note that if we don't use hidden state variables $\mathbf{h}_{t-1}$ in the derivation of the prior distribution, then the prior in (2) becomes independent across snapshots and reduces to the prior of vanilla VGAE.

The inner-product decoder is adopted in VGRNN for the experiments in this paper– $\varphi^{\mathrm{dec}}$ in (3)–to clearly demonstrate the advantages of the stochastic recurrent models for the encoder. Potential extensions with other decoders can be integrated with VGRNN if necessary. More specifically,

$$
p\left(\mathbf{A}^{(t)} \mid \mathbf{Z}^{(t)}\right) = \prod_{i=1}^{N_t} \prod_{j=1}^{N_t} p\left((A_{i,j}^{(t)} \mid \mathbf{z}_i^{(t)}, \mathbf{z}_j^{(t)})\right); \; p\left(A_{i,j}^{(t)} = 1 \mid \mathbf{z}_i^{(t)}, \mathbf{z}_j^{(t)}\right) = \mathrm{sigmoid}\left(\mathbf{z}_i^{(t)}(\mathbf{z}_j^{(t)})^T\right),
$$
$$
\tag{8}
$$

where $\mathbf{z}_i^{(t)}$ corresponds to the embedding representation of node $v_i^{(t)} \in \mathcal{V}^{(t)}$ at time step $t$. Note the generating distribution can also be conditioned on $\mathbf{h}_{t-1}$ if we want to generate $\mathbf{X}^{(t)}$ in addition to the adjacency matrix for other applications. In such cases, $\varphi^{\mathrm{dec}}$ should be a highly flexible neural network instead of a simple inner-product function.

### 3.3 Semi-implicit VGRNN (SI-VGRNN)

To further increase the expressive power of the variational posterior of VGRNN, we introduce a SI-VGRNN dynamic node embedding model. We impose a mixing distributions on the variational distribution parameters in (8) to model the posterior of VGRNN with a semi-implicit hierarchical construction:

$$
\mathbf{Z}^{(t)} \sim q(\mathbf{Z}^{(t)} \mid \boldsymbol{\psi}_t), \qquad \boldsymbol{\psi}_t \sim q_{\boldsymbol{\phi}}(\boldsymbol{\psi}_t \mid \mathbf{A}^{(\leq t)}, \mathbf{X}^{(\leq t)}, \mathbf{Z}^{(<t)}) = q_{\boldsymbol{\phi}}(\boldsymbol{\psi}_t \mid \mathbf{A}^{(t)}, \mathbf{X}^{(t)}, \mathbf{h}_{t-1}). \tag{9}
$$

While the variational distribution $q(\mathbf{Z}^{(t)} \mid \boldsymbol{\psi}_t)$ is required to be explicit, the mixing distribution, $q_{\boldsymbol{\phi}}$, is not subject to such a constraint, leading to considerably flexible $\mathbb{E}_{\boldsymbol{\psi}_t \sim q_{\boldsymbol{\phi}}(\boldsymbol{\psi}_t \mid \mathbf{A}^{(t)}, \mathbf{X}^{(t)}, \mathbf{h}_{t-1})}(q(\mathbf{z}_t \mid \boldsymbol{\psi}_t))$. More specifically, SI-VGRNN draws samples from $q_{\boldsymbol{\phi}}$ by transforming random noise $\boldsymbol{\epsilon}_t$ via a graph neural network, which generally leads to an implicit distribution for $q_{\boldsymbol{\phi}}$.

**Inference.** Under the SI-VGRNN construction, the generation, prior and recurrence models are the same as VGRNN (equations (2) to (5)). We indeed have updated the encoder functions as follows:

$$
\boldsymbol{\ell}_j^{(t)} = \mathrm{GNN}_j(\mathbf{A}^{(t)}, \mathrm{CONCAT}(\mathbf{h}_{t-1}, \boldsymbol{\epsilon}_j^{(t)}, \boldsymbol{\ell}_{j-1}^{(t)})); \; \boldsymbol{\epsilon}_j^{(t)} \sim q_j(\boldsymbol{\epsilon}) \text{ for } j = 1, \ldots, L, \; \boldsymbol{\ell}_0^{(t)} = \varphi_\tau^{\mathbf{x}}\left(\mathbf{X}^{(t)}\right)
$$
$$
\boldsymbol{\mu}_{\mathrm{enc}}^{(t)}(\mathbf{A}^{(t)}, \mathbf{X}^{(t)}, \mathbf{h}_{t-1}) = \mathrm{GNN}_\mu(\mathbf{A}^{(t)}, \boldsymbol{\ell}_L^{(t)}), \quad \boldsymbol{\Sigma}_{\mathrm{enc}}^{(t)}(\mathbf{A}^{(t)}, \mathbf{X}^{(t)}, \mathbf{h}_{t-1}) = \mathrm{GNN}_\Sigma(\mathbf{A}^{(t)}, \boldsymbol{\ell}_L^{(t)}),
$$
$$
q(\mathbf{Z}_i^{(t)} \mid \mathbf{A}^{(t)}, \mathbf{X}^{(t)}, \mathbf{h}_{t-1}, \boldsymbol{\mu}_{i,\mathrm{enc}}^{(t)}, \boldsymbol{\Sigma}_{i,\mathrm{enc}}^{(t)}) = \mathcal{N}(\boldsymbol{\mu}_{i,\mathrm{enc}}^{(t)}(\mathbf{A}^{(t)}, \mathbf{X}^{(t)}, \mathbf{h}_{t-1}), \boldsymbol{\Sigma}_{i,\mathrm{enc}}^{(t)}(\mathbf{A}^{(t)}, \mathbf{X}^{(t)}, \mathbf{h}_{t-1})),
$$

where $L$ is the number of stochastic layers and $\boldsymbol{\epsilon}_j^{(t)}$ is $N_t$-dimensional random noise drawn from a distribution $q_j$ with $N_t$ denoting number of nodes at time $t$. Note that given $\{\mathbf{A}^{(t)}, \mathbf{X}^{(t)}, \mathbf{h}_{t-1}\}$, $\boldsymbol{\mu}_{i,\mathrm{enc}}^{(t)}$ and $\boldsymbol{\Sigma}_{i,\mathrm{enc}}^{(t)}$ are now random variables rather than analytic and thus the posterior is not Gaussian after marginalizing.

Table 1: Dataset statistics.

| Metrics | Enron | COLAB | Facebook | HEP-TH | Cora | Social Evolution |
|---|---|---|---|---|---|---|
| Number of Snapshots | 11 | 10 | 9 | 40 | 11 | 27 |
| Number of Nodes | 184 | 315 | 663 | 1199-7623 | 708-2708 | 84 |
| Number of Edges | 115-266 | 165-308 | 844-1068 | 769-34941 | 406-5278 | 303-1172 |
| Average Density | 0.01284 | 0.00514 | 0.00591 | 0.00117 | 0.00154 | 0.21740 |
| Number of Node Attributes | - | - | - | - | 1433 | 168 |

**Learning.** In this construction, because the parameters of the posterior are random variables, the ELBO goes beyond the simple VGRNN in (7) and can be written as

$$\mathcal{L} = \sum_{t=1}^{T} \Bigg\{ \mathbb{E}_{\boldsymbol{\psi}_t \sim q_{\boldsymbol{\phi}}(\boldsymbol{\psi}_t | \mathbf{A}^{(t)}, \mathbf{X}^{(t)}, \mathbf{h}_{t-1})} \mathbb{E}_{\mathbf{Z}^{(t)} \sim q(\mathbf{Z}^{(t)} | \boldsymbol{\psi}_t)} \log \Big( p(\mathbf{A}^{(t)} | \mathbf{Z}^{(t)}, \mathbf{h}_{t-1}) \Big)$$
$$- \mathbf{KL}\Bigg( \mathbb{E}_{\boldsymbol{\psi}_t \sim q_{\boldsymbol{\phi}}(\boldsymbol{\psi}_t | \mathbf{A}^{(t)}, \mathbf{X}^{(t)}, \mathbf{h}_{t-1})} q\Big(\mathbf{Z}^{(t)} | \boldsymbol{\psi}_t\Big) \,||\, p(\mathbf{Z}^{(t)} | \mathbf{h}_{t-1}) \Big) \Bigg\}. \tag{10}$$

Direct optimization of the ELBO in SIVI is not tractable [25], hence to infer variational parameters of SI-VGRNN, we derive a lower bound for the ELBO as follows (see the supplements for more details.).

$$\underline{\mathcal{L}} = \sum_{t=1}^{T} \mathbb{E}_{\boldsymbol{\psi}_t \sim q_{\boldsymbol{\phi}}(\boldsymbol{\psi}_t | \mathbf{A}^{(t)}, \mathbf{X}^{(t)}, \mathbf{h}_{t-1})} \mathbb{E}_{\mathbf{Z}^{(t)} \sim q(\mathbf{Z}^{(t)} | \boldsymbol{\psi}_t)} \log \left( \frac{p(\mathbf{A}^{(t)} | \mathbf{Z}^{(t)}, \mathbf{h}_{t-1}) \, p(\mathbf{Z}^{(t)} | \mathbf{h}_{t-1})}{q(\mathbf{Z}^{(t)} | \boldsymbol{\psi}_t)} \right). \tag{11}$$

## 4 Experiments

**Datasets.** We evaluate our proposed methods, VGRNN and SI-VGRNN, and baselines on six real-world dynamic graphs as described in Table 1. More detailed descriptions of the datasets can be found in the supplement.

**Competing methods.** We compare the performance of our proposed methods against four competing node embedding methods, three of which have the capability to model evolving graphs with changing node and edge sets. Among these four, two (**DynRNN** and **DynAERNN** [11]) are based on RNN models. By comparing our models to these methods, we will be able to see how much improvement we may obtain by improving the backbone RNN with our new prior construction compared to these RNNs with deterministic hidden states. We also compare our methods against a deep autoencoder with fully connected layers (**DynAE** [11]) to show the advantages of RNN based sequential learning methods. Last but not least, our methods are compared with **VGAE** [14], which is implemented to analyze each snapshot separately, to demonstrate how temporal dependencies captured through hidden states in the backbone GRNN can improve the performance. More detailed descriptions of these selected competing methods are described in the supplements.

**Evaluation tasks.** In the dynamic graph embedding literature, the term *link prediction* has been used with different definitions. While some of the previous works focused on link prediction in a transductive setting and others proposed inductive models, our models are capable of working in both settings. We evaluate our proposed models on three different *link prediction* tasks that have been widely used in the dynamic graph representation learning studies. More specifically, given partially observed snapshots of a dynamic graph $\mathcal{G} = \{G^{(1)}, \ldots, G^{(T)}\}$ with node attributes $\mathcal{X} = \{\mathbf{X}^{(1)}, \ldots, \mathbf{X}^{(T)}\}$, dynamic link prediction problems are defined as follows: 1) **dynamic link detection**, i.e. detect unobserved edges in $G^{(T)}$; 2) **dynamic link prediction**, i.e. predict edges in $G^{(T+1)}$; 3) **dynamic new link prediction**, i.e. predict edges in $G^{(T+1)}$ that are not in $G^{(T)}$.

**Experimental setups.** For performance comparison, we evaluate different methods based on their ability to correctly classify true and false edges. For dynamic link detection problem, we randomly remove 5% and 10% of all edges at each time for validation and test sets, respectively. We also randomly select the equal number of non-links as validation and test sets to compute average precision (AP) and area under the ROC curve (AUC) scores. For dynamic (new) link prediction, all (new) edges are set to be true edges and the same number of non-links are randomly selected to compute AP and AUC scores. In all of our experiments, we test the model on the last three snapshots of dynamic

Table 2: AUC and AP scores of inductive dynamic link detection on dynamic graphs.

| Metrics | Methods | Enron | COLAB | Facebook | Social Evo. | HEP-TH | Cora |
|---|---|---|---|---|---|---|---|
| AUC | VGAE | 88.26 ± 1.33 | 70.49 ± 6.46 | 80.37 ± 0.12 | 79.85 ± 0.85 | 79.31 ± 1.97 | 87.60 ± 0.54 |
| | DynAE | 84.06 ± 3.30 | 66.83 ± 2.62 | 60.71 ± 1.05 | 71.41 ± 0.66 | 63.94 ± 0.18 | 53.71 ± 0.48 |
| | DynRNN | 77.74 ± 5.31 | 68.01 ± 5.50 | 69.77 ± 2.01 | 74.13 ± 1.74 | 72.39 ± 0.63 | 76.09 ± 0.97 |
| | DynAERNN | 91.71 ± 0.94 | 77.38 ± 3.84 | 81.71 ± 1.51 | 78.67 ± 1.07 | 82.01 ± 0.49 | 74.35 ± 0.85 |
| | GRNN | 91.09 ± 0.67 | 86.40 ± 1.48 | 85.60 ± 0.59 | 78.27 ± 0.47 | 89.00 ± 0.46 | 91.35 ± 0.21 |
| | **VGRNN** | **94.41 ± 0.73** | **88.67 ± 1.57** | **88.00 ± 0.57** | **82.69 ± 0.55** | **91.12 ± 0.71** | **92.08 ± 0.35** |
| | **SI-VGRNN** | **95.03 ± 1.07** | **89.15± 1.31** | **88.12 ± 0.83** | **83.36 ± 0.53** | **91.05 ± 0.92** | **94.07 ± 0.44** |
| AP | VGAE | 89.95 ± 1.45 | 73.08 ± 5.70 | 79.80 ± 0.22 | 79.41 ± 1.12 | 81.05 ± 1.53 | 89.61 ± 0.87 |
| | DynAE | 86.30 ± 2.43 | 67.92 ± 2.43 | 60.83 ± 0.94 | 70.18 ± 1.98 | 63.87 ± 0.21 | 53.84 ± 0.51 |
| | DynRNN | 81.85 ± 4.44 | 73.12 ± 3.15 | 70.63 ± 1.75 | 72.15 ± 2.30 | 74.12 ± 0.75 | 76.54 ± 0.66 |
| | DynAERNN | 93.16 ± 0.88 | 83.02 ± 2.59 | 83.36 ± 1.83 | 77.41 ± 1.47 | 85.57 ± 0.93 | 79.34 ± 0.77 |
| | GRNN | 93.47 ± 0.35 | 88.21 ± 1.35 | 84.77 ± 0.62 | 76.93± 0.35 | 89.50 ± 0.42 | 91.37 ± 0.27 |
| | **VGRNN** | **95.17 ± 0.41** | **89.74 ± 1.31** | **87.32 ± 0.60** | **81.41 ± 0.53** | **91.35 ± 0.77** | **92.92 ± 0.28** |
| | **SI-VGRNN** | **96.31 ± 0.72** | **89.90 ± 1.06** | **87.69 ± 0.92** | **83.20± 0.57** | **91.42 ± 0.86** | **94.44 ± 0.52** |

graphs while learning the parameters of the models based on the rest of the snapshots except for HEP-TH where we test the model on the last 10 snapshots. For the datasets without node attributes, we consider the $N_t$-dimensional identity matrix as node attributes at time $t$. Numbers show mean results and standard error for 10 runs on random datasets splits with random initializations.

For all datasets, we set up our VGRNN model to have a single recurrent hidden layer with 32 GRU units. All $\varphi$'s in equations (3), (4), and (6) are modeled by a 32-dimensional fully-connected layer. We use two 32-dimensional fully-connected layers for $\varphi^{\text{prior}}$ in (2) and 2-layer GCN with sizes equal to [32, 16] to model $\boldsymbol{\mu}_{\text{enc}}^{(t)}$ and $\boldsymbol{\sigma}_{\text{enc}}^{(t)}$ in (6). For SI-VGRNN, a stochastic GCN layer with size 32 and an additional GCN layer of size 16 are used to model the $\boldsymbol{\mu}$. The dimension of injected standard Gaussian noise $\epsilon$ is 16. The covariance matrix $\boldsymbol{\Sigma}$ is deterministic and is inferred through two layers of GCNs with sizes equal to [32, 16]. For fair comparison, the number of parameters are the same for the competing methods. In all experiments, we train the models for 1500 epochs with the learning rate 0.01. We use the validation set for the early stopping. The supplement contains additional implementation details with hyperparmaeter selection. We implemented (SI-)VGRNN in PyTorch [18] and the implementation of our proposed models is accessible at `https://github.com/VGraphRNN/VGRNN`.

## 4.1 Results and discussion

**Dynamic link detection.** Table 2 summarizes the results for inductive link detection in different datasets. Our proposed methods, VGRNN and SI-VGRNN, outperform competing methods across all datasets by large margins. Improvement made by (SI-)VGRNN compared to GRNN and DynAERNN supports our claim that latent random variables carry more information than deterministic hidden states specially for dynamic graphs with complex temporal changes. Comparing the (SI-)VGRNN with VGAE, which is a static graph embedding method, shows that the improvement of the proposed methods is not only because of introducing stochastic latent variables, but also successful modelling of temporal dependencies. We note that methods that take node attributes as input, i.e VGAE, GRNN and (SI-)VGRNN, outperform other competing methods by a larger margin in Cora dataset which includes node attributes.

Comparing SI-VGRNN with VGRNN shows that the Gaussian latent distribution may not always be the best choice for latent node representations. SI-VGRNN with flexible variational inference can learn more complex latent structures. The results for the Cora dataset, which also includes attributes, clearly magnify the benefits of flexible posterior as SI-VGRNN improves the accuracy by 2% compared to VGRNN. We also note that the improvement made by SI-VGRNN compared to VGRNN is marginal in Facebook dataset. The reason could be that Gaussian latent variables already represent the graph well. Therefore, more flexible posteriors do not enhance the performance significantly.

**Dynamic (new) link prediction.** Tables 3 and 4 show the results for link prediction and new link prediction, respectively. Since GRNN is trained as an autoencoder, it cannot predict edges in the next snapshot. However, in (SI-)VGRNN, the prior construction based on previous time steps allows us to predict links in the future. Note that none of the methods can predict new nodes, therefore, HEP-TH, Cora and Citeseer datasets are not evaluated for these tasks. VGRNN and SI-VGRNN outperform the competing methods significantly in both tasks for all of the datasets which proves

Table 3: AUC and AP scores of dynamic link prediction on real-world dynamic graphs.

| Metrics | Methods | Enron | COLAB | Facebook | Social Evo. |
|---------|---------|-------|-------|----------|-------------|
| **AUC** | DynAE | $74.22 \pm 0.74$ | $63.14 \pm 1.30$ | $56.06 \pm 0.29$ | $65.50 \pm 1.66$ |
| | DynRNN | $86.41 \pm 1.36$ | $75.7 \pm 1.09$ | $73.18 \pm 0.60$ | $71.37 \pm 0.72$ |
| | DynAERNN | $87.43 \pm 1.19$ | $76.06 \pm 1.08$ | $76.02 \pm 0.88$ | $73.47 \pm 0.49$ |
| | **VGRNN** | $\mathbf{93.10 \pm 0.57}$ | $\mathbf{85.95 \pm 0.49}$ | $89.47 \pm 0.37$ | $77.54 \pm 1.04$ |
| | **SI-VGRNN** | $\mathbf{93.93 \pm 1.03}$ | $\mathbf{85.45 \pm 0.91}$ | $\mathbf{90.94 \pm 0.37}$ | $\mathbf{77.84 \pm 0.79}$ |
| **AP** | DynAE | $76.00 \pm 0.77$ | $64.02 \pm 1.08$ | $56.04 \pm 0.37$ | $63.66 \pm 2.27$ |
| | DynRNN | $85.61 \pm 1.46$ | $78.95 \pm 1.55$ | $75.88 \pm 0.42$ | $69.02 \pm 1.71$ |
| | DynAERNN | $89.37 \pm 1.17$ | $81.84 \pm 0.89$ | $78.55 \pm 0.73$ | $71.79 \pm 0.81$ |
| | **VGRNN** | $\mathbf{93.29 \pm 0.69}$ | $\mathbf{87.77 \pm 0.79}$ | $89.04 \pm 0.33$ | $77.03 \pm 0.83$ |
| | **SI-VGRNN** | $\mathbf{94.44 \pm 0.85}$ | $\mathbf{88.36 \pm 0.73}$ | $\mathbf{90.19 \pm 0.27}$ | $\mathbf{77.40 \pm 0.43}$ |

Table 4: AUC and AP scores of dynamic new link prediction on real-world dynamic graphs.

| Metrics | Methods | Enron | COLAB | Facebook | Social Evo. |
|---------|---------|-------|-------|----------|-------------|
| **AUC** | DynAE | $66.10 \pm 0.71$ | $58.14 \pm 1.16$ | $54.62 \pm 0.22$ | $55.25 \pm 1.34$ |
| | DynRNN | $83.20 \pm 1.01$ | $71.71 \pm 0.73$ | $73.32 \pm 0.60$ | $65.69 \pm 3.11$ |
| | DynAERNN | $83.77 \pm 1.65$ | $71.99 \pm 1.04$ | $76.35 \pm 0.50$ | $66.61 \pm 2.18$ |
| | **VGRNN** | $\mathbf{88.43 \pm 0.75}$ | $\mathbf{77.09 \pm 0.23}$ | $87.20 \pm 0.43$ | $\mathbf{75.00 \pm 0.97}$ |
| | **SI-VGRNN** | $\mathbf{88.60 \pm 0.95}$ | $\mathbf{77.95 \pm 0.41}$ | $\mathbf{87.74 \pm 0.53}$ | $76.45 \pm 1.19$ |
| **AP** | DynAE | $66.50 \pm 1.12$ | $58.82 \pm 1.06$ | $54.57 \pm 0.20$ | $54.05 \pm 1.63$ |
| | DynRNN | $80.96 \pm 1.37$ | $75.34 \pm 0.67$ | $75.52 \pm 0.50$ | $63.47 \pm 2.70$ |
| | DynAERNN | $85.16 \pm 1.04$ | $77.68 \pm 0.66$ | $78.70 \pm 0.44$ | $65.03 \pm 1.74$ |
| | **VGRNN** | $\mathbf{87.57 \pm 0.57}$ | $\mathbf{79.63 \pm 0.94}$ | $86.30 \pm 0.29$ | $73.48 \pm 1.11$ |
| | **SI-VGRNN** | $\mathbf{87.88 \pm 0.84}$ | $\mathbf{81.26 \pm 0.38}$ | $\mathbf{86.72 \pm 0.54}$ | $\mathbf{73.85 \pm 1.33}$ |

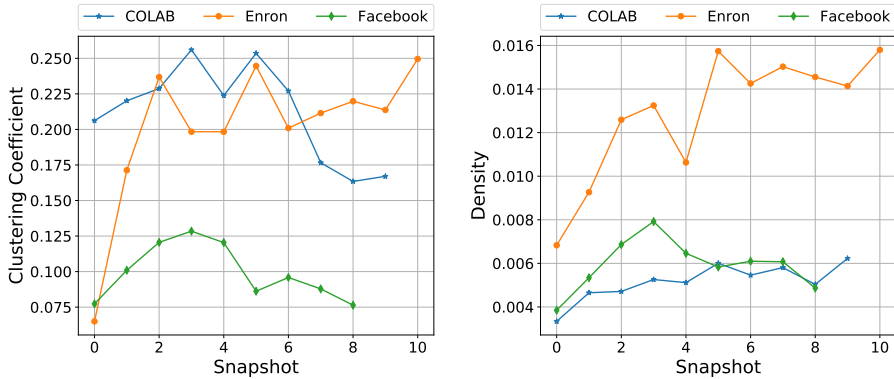

Figure 2: Evolution of graph statistics through time.

that our proposed models have better generalization, which is the result of including random latent variables in our model. We note that our proposed methods improve new link prediction more substantially which shows that they can capture temporal trends better than the competing methods. Comparing VGRNN with SI-VGRNN shows that the prediction results are almost the same for all datasets. The reason is that although the posterior is more flexible in SI-VGRNN, the prior on which our predictions are based, is still Gaussian, hence the improvement is marginal. A possible avenue for further improvements is constructing more flexible priors such as semi-implicit priors proposed by Molchanov et al. [17], which we leave for future studies.

To find out when VGRNN and SI-VGRNN show more improvements compared to the baselines, we take a closer look at three of the datasets. Figure 2 shows the temporal evolution of density and clustering coefficients of COLAB, Enron, and Facebook datasets. Enron shows the highest density and clustering coefficients, indicating that it contains dense clusters who are densely connected with each other. COLAB have low density and high clustering coefficients across time, which means that although it is very sparse but edges are mostly within the clusters. Facebook, which has both low density and clustering coefficients, is very sparse with almost no clusters. Looking back at (new) link prediction results, we see that the improvement margin of (SI-)VGRNN compared to competing methods is more substantial for Facebook. Moreover, the improvement margin diminishes when the graph has more clusters and is more dense. Predicting the evolution very sparse graphs with no clusters is indeed a very difficult task (arguably more difficult than dense graphs), in which our proposed (SI-)VGRNN is very successful. The stochastic latent variables in our models can capture the temporal trend while other methods tend to overfit very few observed links.

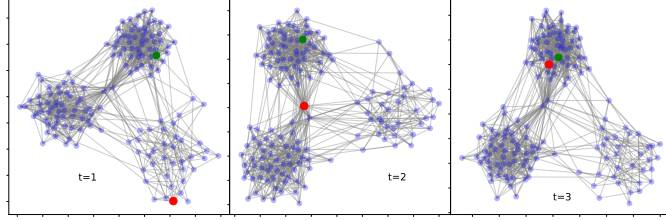

Figure 3: Evolution of simulated graph topology through time.

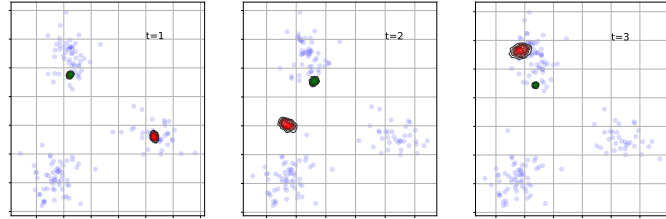

Figure 4: Latent representations of the simulated graph in different time steps in 2-d space using VGRNN.

## 4.2 Interpretable latent representations

To show that VGRNN learns more interpretable latent representations, we simulated a dynamic graph with three communities in which a node (red colored node) transfers from one community into another in two time steps (Figure 3). We embedded the node into 2-d latent space using VGRNN (Figure 4) and DynAERNN (the best performed baseline; Figure S1 in the supplementary material). While the advantages of modeling uncertainty for latent representations and its relation to node labels (classes) for static graphs have been discussed in Bojchevski and Günnemann [2], we argue that the uncertainty is also directly related to topological evolution in dynamic graphs.

More specifically, the variance of the latent variables for the node of interest increases in time (left to right) marked with the red contour. In time steps 2 and 3 (where the node is moving in the graph), the information from previous and current time steps contradicts each other; hence we expect the representation uncertainty to increase. We also plotted the variance of a node whose community doesn't change in time (marked with the green contour). As we expected, the variance of this node does not increase over time. We argue that the uncertainty helps to better encode non-smooth evolution, in particular abrupt changes, in dynamic graphs. Moreover, at time step 2, the moving node have multiple edges with nodes in two communities. Considering the inner-product decoder, which is based on the angle between the latent representations, the moving node can be connected to both of the communities which is consistent with the graph topology. We note that DynAERNN (Figure S1) fails to produce such an interpretable latent representation. We can see that VGRNN can separate the communities in the latent space more distinctively than what DynAERNN does.

## 5 Conclusion

We have proposed VGRNN and SI-VGRNN, the first node embedding methods for dynamic graphs that embed each node to a random vector in the latent space. We argue that adding high level latent variables to graph recurrent neural networks not only increases its expressiveness to better model the complex dynamics of graphs, but also generates interpretable random latent representation for nodes. SI-VGRNN is also developed by combining VGRNN and semi-implicit variational inference for flexible variational inference. We have tested our proposed methods on dynamic link prediction tasks and they outperform competing methods substantially, specially for very sparse graphs.

## 6 Acknowledgments

The presented materials are based upon the research supported by the National Science Foundation under Grants ENG-1839816, IIS-1848596, CCF-1553281, IIS-1812641 and IIS-1812699. We also thank Texas A&M High Performance Research Computing and Texas Advanced Computing Center for providing computational resources to perform experiments in this work.

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
