[Supplementary Material]

# Variational Graph Recurrent Neural Networks: Supplementary Material

**Ehsan Hajiramezanali**[†*], **Arman Hasanzadeh**[†*], **Nick Duffield**[†], **Krishna Narayanan**[†],
**Mingyuan Zhou**[‡], **Xiaoning Qian**[†]

† Department of Electrical and Computer Engineering, Texas A&M University
{ehsanr, armanihm, duffieldng, krn, xqian}@tamu.edu
‡ McCombs School of Business, The University of Texas at Austin
mingyuan.zhou@mccombs.utexas.edu

This document contains the detailed discussion of related works, the derivation of the ELBO lower bound for SI-VGRNN inference, additional dataset details, experimental setups and implementation details as well as additional results on interpretability of the derived latent representations.

## A  Related works

Several dynamic graph embedding methods have been developed using various techniques such as matrix factorization [23, 21], random walk [20, 7], deep learning [13, 2, 3, 10], and stochastic process [22, 14, 15]. The shortcomings of the existing methods can be categorized as follows:

- Most of these existing methods either capture topological evolution or node attribute changes to learn dynamic node embeddings [18, 11]. But only a few of them model both changes simultaneously [15].

- Some of the existing methods, such as the ones in [22, 2, 21], assume that the temporal patterns of evolving processes are of short duration and fail to capture long-range temporal dependencies in dynamic networks.

- A common assumption in the literature is that the topological changes are smooth. The methods with this assumption [2, 21] usually use a regularization term to avoid abrupt changes, which limits their flexibility. Deep learning based models, such as the ones in [10, 3], have been proposed to address this shortcoming; however, these methods only care about the topological changes over time but do not model node attribute dynamics or complex dependencies between two evolving processes.

- Many of the existing methods, such as [13, 15], cannot model the deletion of nodes or edges which limits their generalizability and flexibility.

- While generative models in form of parametric temporal point processes [14] and deep temporal point processes [15] have been used for modeling dynamic graphs, none of the existing methods are capable of modeling the uncertainty of the latent representations.

Our proposed (SI-)VGRNN is the first variational based deep generative model for representation learning of dynamic graphs. On the contrary to existing methods, (SI-)VGRNN is capable of inferring the uncertainty of latent representations which is the key in modeling non-smooth changes in dynamic graphs. Moreover, (SI-)VGRNN can capture long-term dependencies in node attribute dynamics as well as topological evolution. Furthermore, (SI-)VGRNN can handle node and edge addition/deletion.

---

[*]Both authors contributed equally.

## B  Lower bound for ELBO in SI-VGRNN

SI-VGRNN posterior can be derived by marginalizing out the mixing distribution as follows,

$$\mathbf{Z}^{(t)} \sim q(\mathbf{Z}^{(t)} \,|\, \psi_t), \qquad \psi_t \sim q_\phi(\psi_t \,|\, \mathbf{A}^{(\leq t)}, \mathbf{X}^{(\leq t)}, \mathbf{Z}^{(<t)}) = q_\phi(\psi_t | \mathbf{A}^{(t)}, \mathbf{X}^{(t)}, \mathbf{h}_{t-1}),$$

$$g_\phi(\mathbf{Z}^{(t)} \,|\, \mathbf{A}^{(t)}, \mathbf{X}^{(t)}, \mathbf{h}_{t-1}) = \int_{\psi_t} q(\mathbf{Z}^{(t)} \,|\, \psi_t) \, q_\phi(\psi_t \,|\, \mathbf{A}^{(t)}, \mathbf{X}^{(t)}, \mathbf{h}_{t-1}) \, d\psi_t \,.$$

Based on the first theorem in Yin and Zhou [19], which shows that

$$\mathbf{KL}(\mathbb{E}_{\psi \sim q_\phi(\psi \,|\, \mathbf{X}, \mathbf{A})}[q(\mathbf{Z} \,|\, \psi)] \,||\, p(\mathbf{Z})) \leq \mathbb{E}_{\psi \sim q_\phi(\psi \,|\, \mathbf{X}, \mathbf{A})}[\mathbf{KL}(q(\mathbf{Z} \,|\, \psi) \,||\, p(\mathbf{Z}))],$$

the lower bound for ELBO can be derived as follows,

$$
\begin{aligned}
\underline{\mathcal{L}} &= \sum_{t=1}^{T} \underline{\mathcal{L}}\left( q(\mathbf{Z}^{(t)} \,|\, \psi_t), \, q_\phi(\psi_t \,|\, \mathbf{A}^{(t)}, \mathbf{X}^{(t)}, \mathbf{h}_{t-1}) \right) \\
&= \sum_{t=1}^{T} \mathbb{E}_{\psi_t \sim q_\phi(\psi_t \,|\, \mathbf{A}^{(t)}, \mathbf{X}^{(t)}, \mathbf{h}_{t-1})} \mathbb{E}_{\mathbf{Z}^{(t)} \sim q(\mathbf{Z}^{(t)} \,|\, \psi_t)} \log \left( \frac{p(\mathbf{A}^{(t)} \,|\, \mathbf{Z}^{(t)}, \mathbf{h}_{t-1}) \, p(\mathbf{Z}^{(t)} \,|\, \mathbf{h}_{t-1})}{q(\mathbf{Z}^{(t)} \,|\, \psi_t)} \right) \\
&= -\sum_{t=1}^{T} \mathbb{E}_{\psi_t \sim q_\phi(\psi_t \,|\, \mathbf{A}^{(t)}, \mathbf{X}^{(t)}, \mathbf{h}_{t-1})} \mathbf{KL}\left( q(\mathbf{Z}^{(t)} \,|\, \psi_t) \,||\, p(\mathbf{Z}^{(t)} \,|\, \mathbf{h}_{t-1}) \right) \\
&\qquad + \mathbb{E}_{\psi_t \sim q_\phi(\psi_t \,|\, \mathbf{A}^{(t)}, \mathbf{X}^{(t)}, \mathbf{h}_{t-1})} \mathbb{E}_{\mathbf{Z}^{(t)} \sim q(\mathbf{Z}^{(t)} \,|\, \psi_t)} \log p(\mathbf{A}^{(t)} \,|\, \mathbf{Z}^{(t)}, \mathbf{h}_{t-1}) \\
&\leq -\sum_{t=1}^{T} \mathbf{KL}\left( \mathbb{E}_{\psi_t \sim q_\phi(\psi_t \,|\, \mathbf{A}^{(t)}, \mathbf{X}^{(t)}, \mathbf{h}_{t-1})} q(\mathbf{Z}^{(t)} \,|\, \psi_t) \,||\, p(\mathbf{Z}^{(t)} \,|\, \mathbf{h}_{t-1}) \right) \\
&\qquad + \mathbb{E}_{\psi_t \sim q_\phi(\psi_t \,|\, \mathbf{A}^{(t)}, \mathbf{X}^{(t)}, \mathbf{h}_{t-1})} \mathbb{E}_{\mathbf{Z}^{(t)} \sim q(\mathbf{Z}^{(t)} \,|\, \psi_t)} \log p(\mathbf{A}^{(t)} \,|\, \mathbf{Z}^{(t)}, \mathbf{h}_{t-1}) \\
&= \sum_{t=1}^{T} \mathbb{E}_{\mathbf{Z}^{(t)} \sim g_\phi(\mathbf{Z}^{(t)} \,|\, \mathbf{A}^{(t)}, \mathbf{X}^{(t)}, \mathbf{h}_{t-1})} \log \left( \frac{p(\mathbf{A}^{(t)} \,|\, \mathbf{Z}^{(t)}, \mathbf{h}_{t-1}) \, p(\mathbf{Z}^{(t)} \,|\, \mathbf{h}_{t-1})}{g_\phi(\mathbf{Z}^{(t)} \,|\, \mathbf{A}^{(t)}, \mathbf{X}^{(t)}, \mathbf{h}_{t-1})} \right) \\
&= \mathbb{E}_{\mathbf{Z} \sim q(\mathbf{Z}^{(\leq t)} \,|\, \mathbf{A}^{(\leq t)}, \mathbf{X}^{(\leq t)})} \left[ \log p(\mathbf{A}^{(\leq t)}, \mathbf{X}^{(\leq t)}, \mathbf{Z}^{(\leq t)}) - \log q(\mathbf{Z}^{(\leq t)} \,|\, \mathbf{A}^{(\leq t)}, \mathbf{X}^{(\leq t)}) \right] \\
&= \mathcal{L}
\end{aligned}
$$

While a Monte Carlo estimation of $\underline{\mathcal{L}}$ only requires $q_\phi(\mathbf{Z}^{(t)} \,|\, \psi_t)$ to have an analytic density functions and $q_\phi(\psi_t \,|\, \mathbf{X}^{(t)}, \mathbf{h}_{t-1})$ to be convenient to sample from, the marginal posterior $g_\phi(\mathbf{Z}^{(t)} \,|\, \mathbf{X}^{(t)}, \mathbf{h}_{t-1})$ is often intractable and so the Monte Carlo estimation of the ELBO $\mathcal{L}$ is prohibited. SI-VGRNN evaluates the lower bound separately from the distribution sampling. This captures the idea that combining an explicit $q_\phi(\mathbf{Z}^{(t)} \,|\, \psi_t)$ with an implicit $q_\phi(\psi_t \,|\, \mathbf{X}^{(t)}, \mathbf{h}_{t-1})$ is as powerful as needed, but makes the computation tractable.

As discussed in [19], if optimizing the variational parameter by climbing $\underline{\mathcal{L}}$, without stopping the optimization algorithm early, $q_\phi(\psi_t \,|\, \mathbf{X}^{(t)}, \mathbf{h}_{t-1})$ could converge to a point mass density, making SI-VGRNN degenerate to VGRNN. To prevent this problem and inspired by SIVI, we add a regularization term to the lower bound as follows,

$$\underline{\mathcal{L}}_K = \underline{\mathcal{L}} + B_K,$$

where

$$B_K = \sum_{t=1}^{T} \mathbb{E}_{\psi_t, \psi_t^{(1)}, \dots, \psi_t^{(K)} \sim q_\phi(\psi_t \,|\, \mathbf{A}^{(t)}, \mathbf{X}^{(t)}, \mathbf{h}_{t-1})} \mathbf{KL}(q(\mathbf{Z}^{(t)} \,|\, \psi_t) \,||\, \tilde{g}_K(\mathbf{Z}^{(t)} | \mathbf{A}^{(t)}, \mathbf{X}^{(t)}, \mathbf{h}_{t-1})),$$

$$\tilde{g}_K(\mathbf{Z}^{(t)} \,|\, \mathbf{A}^{(t)}, \mathbf{X}^{(t)}, \mathbf{h}_{t-1})) = \frac{q_\phi(\psi_t \,|\, \mathbf{A}^{(t)}, \mathbf{X}^{(t)}, \mathbf{h}_{t-1}) + \sum_{k=1}^{K} q_\phi(\psi_t^{(k)} \,|\, \mathbf{A}^{(t)}, \mathbf{X}^{(t)}, \mathbf{h}_{t-1})}{K + 1}.$$

The lower bound leads to an asymptotically exact ELBO that satisfies $\underline{\mathcal{L}}_0 = \underline{\mathcal{L}}$ and $\lim_{K \to \infty} \underline{\mathcal{L}}_K = \mathcal{L}$.

## C   Additional dataset details

**Enron emails (Enron).** This graph constructed from 500,000 email messages exchanged between 184 Enron employees from 1998 to 2002 [8]. The nodes represent the employees and the edges are emails exchanged between two employees. Following the same procedure as in [17, 9] we clean the data to get 10 temporal snapshots of the graph. This graph does not have any node or edge attribute.

**Collaboration (COLAB).** This dataset represents collaborations between 315 authors. Each node in this dynamic graph is an author and the edges represent co-authorship relationships. The data, provided by Rahman and Al Hasan [9], are collected from years 2000-2009 with a total of 10 snapshots considering each year as a time stamp. This COLAB graph does not have any node or edge attribute.

**Facebook.** The Facebook wall posts dynamic graph, provided by [16], has 9 time stamps. Following the same data cleaning procedure as in [17, 9], we get 663 nodes at each snapshot. No node or edge attribute is provided for this graph.

**HEP-TH.** The original dataset [1] covers all the citations of the papers in High Energy Physics Theory conference from January 1993 to April 2003 [5]. For each month, we create a citation graph using all the papers published up to that month. We only consider the first ten months leading to 10 snapshots in this dynamic graph. The graph has 1199 nodes at the first month and 2462 at the last one. This graph also has no node or edge attributes.

**Cora.** The Cora dataset is another citation graph consists of 2708 scientific publications [12]. The nodes in the graph represent the publications and the edges indicate the citation relations. Each node is provided with a 1433-dimensional binary attribute vector. Each dimension of the attribute vector indicates the presence of a word in the publication from a dictionary. Originally, Cora is a static graph dataset, therefore in order to use it in a dynamic fashion, we preprocess the data as follows in the same manner as in [6]. We take the indices of nodes as their arriving order in the dynamic graph and add 200 nodes with their corresponding edges, at each temporal snapshot. The dynamic graph includes 11 snapshots, starting with 708 nodes and reaches to 2708 nodes at the last snapshot.

**Social evolution.** The social evolution dataset is collected from Jan 2008 to June 30, 2009 and released by MIT Human Dynamics Lab [15]. For this dataset, we consider Calls and SMS records between users as node attributes and all Close Friendship records and Proximity as graph topology. We consider the collected information from Jan 2008 until Sep 10, 2008 (i.e. survey date) to form the initial network. We used cumulative data for 10 days periods of to form a snapshot of dynamic network for 27 snapshots.

## D   Details on the experimental setup and hyper-parameters selection

**Dynamic autoencoder (DynAE)** [3]. This autoencoder model uses multiple fully connected layers for both encoder and decoder to capture highly non-linear interactions between nodes at each time step and across multiple time steps. It can take a set of graphs with different adjacency matrices. This model has $\mathcal{O}(nld_1)$ parameters, where $n$, $l$, and $d_1$ are the number of nodes, autoregressive lag, and dimension of the first hidden layer, respectively. Learning to optimize this huge number of parameters can be challenging for sparse graphs [3], which is often the case when studying real-world datasets. The input to this model at each node is the neighborhood vector of that node.

**Dynamic recurrent neural network (DynRNN)** [3]. This model uses LSTM networks as both encoder and decoder to capture the long-term dependencies in dynamic graphs. Comparing to DynAE, the number of parameters is reduced and the model is capable of learning complex temporal patterns more efficiently. The input to this model at each node is the neighborhood vector of that node.

**Dynamic autoenncoder recurrent neural network (DynAERNN)** [3]. Instead of passing the input adjacency matrices into LSTM, DynAERNN uses a fully connected encoder to initially acquire low dimensional hidden representations and then pass them as the input of LSTM to learn the embedding. The decoder of this model is a fully connected network similar to DynAE. The input to this model at each node is the neighborhood vector of that node.

**Experimental setups.**   For VGAE at each snapshot, we use two GCN layers with 32 and 16 units for $\mathbf{GCN}_\mu$ and $\mathbf{GCN}_\sigma$. Since VGAE is a method for static graph embedding, we start training with

Figure S1: Latent representation of the simulated graph in different time steps in 2-d space using DynAERNN.

the first snapshot and use the inferred parameters as initialization for the next snapshot. We continue this process until the last training snapshot. In all VGAE experiments, the learning rate is set to be 0.01. We learn the model for 500 training epochs and use the validation set for the early stopping. We use the code provided by the author [4] in our experiments. For DynAE, DynRNN, and DynAERNN, we chose the dimension and number of layers of the encoder and decoder such that the total numbers of parameters is comparable to (SI-)VGRNN. For these methods, we use the source code published by the authors. In these methods, the learning rate is set to be 0.01 and the learning procedure converges in 250 training epochs. The *look back* parameter in these models, which indicates how much in the past the model looks to learn the embedding, is set to be 2. In all of the experiments in this paper, the embedding dimension is set to 16 except for HEP-TH where embedding dimension is 32.

All of the node embedding methods for link prediction performance comparison are run on a single cluster node with dual-GPU Tesla K80 accelerator and 128GB RAM. For running each epoch on the HEP-TH dataset using one of the GPUs on this cluster, SI-VGRNN, VGRNN, DynRNN, DynAERNN, and DynAE take around 36, 12, 40, 5, and 1 seconds, respectively. This is expected as DynRNN has two 2-layer LSTMs as decoders and encoders. On the other hand, the number of parameters in DynAERNN, which includes just one 2-layer LSTM, is less than that of DynRNN. DynAE are faster as they do not have LSTM units.

## E   Additional experimental results on interpretability of latent representations

Here, we include the latent representations of the simulated graph (in Section 4.2 of the main text) learned by DynAERNN (shown in Figure S1). Compared to the latent representation learned by VGRNN, not only DynAERNN is not capable of modeling uncertainty of representations, but also it fails to separate the communities of the graph at different time steps, which VGRNN has successfully accomplished.