[Reviews · NeurIPS 2019]

Reviewer 1



This paper studies a Graph RNN model for dynamic graphs. Cardinalities of nodes and edges can be time-varying. Especially the proposed VGRNN is made for highly variable graph sequences. The hidden state h_t, which is tracked via RNN function, governs the prior of latent variables and the sampled latent variable controls the generation of time-varying adjacency matrices. Such hierarchical modeling allows the proposed VGNN to fit to highly time-variable graph sequences. I had been working on modeling dynamic graphs for years. Modeling is rigorous but somewhat complicated. It forces me to read the entire paper multiple times for understanding. Possibly very difficult for non-expert readers to understand the core formulation. I have two major concerns about this paper. As written in L96, the proposed model is flexible enough to allow the cardinality of node and edge sets change across time steps. In such a dynamic graph, deletions and additions of nodes are always problematic for formulations and implementations. How does the VGRNN model deal with such issues? Unfortunately, I cannot find explanations concerning these issues in the current manuscript. For example, assume at t=0 there are 5 nodes in the graph. Then, consider the 3rd node disappears at t=1, and the 6th (new) node appears at t=2. In such a situation, an appropriate model needs to guarantees that the deleted 3rd node does not affect the computation after t=1, and the added 6th node does not affect the inference before t=2. At the same time, the sizes of adjacency matrix may change over time so it is not trivial to maintain identities of nodes. It is great if the authors can detail how to resolve these issues. Another concern is insufficient discussions about the existing works. In the main manuscript, the discussions about the existing works are limited to the ones very close or direct competitors (there are some discussions in the supplementary material, though). I do understand the tight page limitation of NuerIPS submissions, but the current main manuscript is obviously not sufficient to place the proposed work in the dynamic graph analysis literature. The reference mainly focuses on the node embedding approach, but the scope of dynamic graph analysis is broader. For example, node embedding is not the sole approach for dynamic link detection/prediction problems, right? Why this paper focuses on the node embedding approach? Pros and cons of node embedding, aginst other approaches (such as classical, non-DNN approaches)? + Governing the prior parameters by latent RNN factors sounds reasonable for modeling highly-variable dynamic graphs + Experimental performance seems nice - Not clear how the VGRNN treats the dynamic addition/deletion of nodes - Insufficient discussions with related works ### after author feedback ### I read the feedback letter and other reviewers' questions. In the letter L13 says "the size of A and X can in time but their latent space maintains across time." This partially addressed my concern conceptually, but raised another practical question. If the size of A and X are t-dependent, some components in your model formulation are difficult to implement because they accept size-variable matrices. Especially, Eq.(1), Eq.(4), \mu_{enc}^{(t)} and \sigma_{enc}^{(t)} in Eq. (6) should be described more concretely. Thus I determined to keep my score.

Reviewer 2



Clarity: There exists some typos, e.g., ''graph convolutional neural network (GCRN)'' in line 34. Please proofread the paper. Originality: Though RNN with stochastic latent variables and GRNN has been studied extensively, there is little work combing these two ideas to the best of my knowledge. Quality: This paper claims that high level latent random variables of VGRNN are vital (i.e., interpretability and uncertainty) for modeling complex dynamic graph. Experiment results of AUC and AP support the claim partially. However, there are no experiments discussing the interpretability of latent variables. Significance: VGRNN does contribute to understanding and solving dynamic graph modeling compared to previous methods. ---------------------------------- After Rebuttal: Thanks for your response. The visualization of latent space addresses my question partially.

Reviewer 3



Originality: the effort to predict links in dynamic graphs through a combination of variational autoencoder, graph neural network and recurrent neural network is the interesting contribution of the paper. I have some doubts about the formulation that subtracts from my evaluation of originality. - As I understand the paper, X represents node attributes, A represents dynamic network, Z represents the code of all X and A up to the current time, and h represents a latent state of all X, A and Z up to the current time. Then what additional information does Z_t provide in Eq. 4, since in Eq. 2 Z_t contains the same amount of information as h_{t-1}? - In Fig. 1 (b), the code Z_t is only used to generate the dynamic network A_t, not the node attributes X_t. This is not symmetric if we consider that Z_t encodes both X_t and A_t in Fig. 1. - In the formulation, Z_t has the same dimensionality as X_t, meaning that Z_t is a code for each node. This doesn't necessarily need to be so. - In the experiment, I would like to see more detailed analysis of what GCRN gives us, in comparison with other neural network algorithms, as well as non-neural-network algorithms. Significance: This paper has significance as an application paper. But for application paper, I would like to see deeper analysis of what GCRN gives us in comparison with other algorithms. In particular, some visualization and comparison at the link level might be very helpful. The variational solution of evidence lower bound is pretty standard. ### Thanks for answering the key questions in the serious rebuttal! While I am still not totally convinced about novelty, I believe the authors will seriously revise of the submission and address reviewers' concerns if it is accepted. As a result, I would move my overall score to 6.

[Author Response · NeurIPS 2019]

We truly appreciate helpful comments from all three reviewers. Our main modeling and methodological contributions are: 1) A novel generative model, (SI-)VGRNN, is proposed to achieve more interpretable latent representations for dynamic graphs as shown below. To the best of our knowledge, this is the first method modeling uncertainty of node latent representations for dynamic graphs, capturing both topological evolution and dynamic attribute changes simultaneously. 2) By imposing semi-implicit variational inference, we have further extended our original VGRNN model to increase the expressive power of the inferred posterior. 3) Unlike existing dynamic graph models focusing on specific tasks including link prediction and community detection [Kim et al., 2017], (SI-)VGRNN facilitates end-to-end learning of universal latent representations for various graph analytic tasks.

**R1** asked how (SI-)VGRNN deals with deletions and additions of nodes. If the graph is growing with addition of new nodes, we assume that the prior of latent representations for the newly observed nodes is zero mean with unit variance

Gaussian distribution. If node deletion occurs, we assume that the identity of nodes can be maintained thus removing a node is equivalent to removing all the edges connected to it. More specifically, the sizes of $\mathbf{A}$ and $\mathbf{X}$ can change in time while their latent space maintains across time. Note our model is not designed to predict the occurrence of new nodes.

To show that VGRNN learns more interpretable latent representations (**R1, R3, R4**), we simulated a dynamic graph with three communities in which a node (red) transfers from one community into another in two time steps (1st Fig.). We embedded the node into 2-d latent space using VGRNN (2nd Fig.) and DynAERNN (the best performed baseline; 3rd Fig.). While the advantages of modeling uncertainty for latent representations and its relation to node labels (classes) for static graphs have been discussed in Bojchevski & Gunnemann [2018], we argue that the uncertainty is also directly related to structural evolution of nodes in dynamic graphs.

More specifically, the variance of the latent variables for the desired node increases in time (left to right) colored with red contour. In time steps 2 and 3 (where the node is moving in the graph), the information from previous and current time contradicts each other; hence we expect the representation uncertainty to increase. We also plotted the variance of a node whose community doesn't change in time (colored with green contour). As we expected, the variance of this node does not increase over time. We argue that the uncertainty helps to better encode non-smooth evolution, in particular abrupt changes, in dynamic graphs. Moreover, at time step 2, the moving node have multiple edges with nodes in two communities. Considering the inner-product decoder, which is based on the angle between the latent representations, the moving node can be connected to both of the communities which is consistent with the graph topology. We note that DynAERNN fails to produce such an interpretable latent representation. We can also see that VGRNN can separate the communities in the latent space more distinctively than DynAERNN.

**R4** asked what additional information $\mathbf{Z}_t$ provides in Eq. 4: While Eq. 2 constructs the "prior" distribution for $\mathbf{Z}_t$, as conditioned on the state variable $\mathbf{h}_{t-1}$, the posterior of $\mathbf{Z}_t$ has been fed to $\mathbf{h}_t$ in recurrence step, i.e. Eq. 4. Note that the posterior of $\mathbf{Z}_t$ has been inferred based on the information of $\mathbf{A}_t$, $\mathbf{X}_t$ and $\mathbf{h}_{t-1}$, i.e. Eq. 6. From this point of

view, the information of $\mathbf{Z}_t$ is more than $\mathbf{h}_{t-1}$. We have to feed $\mathbf{h}_{t-1}$ in Eq. 4 to maintain the RNN structure.

**R4** also asked about reconstructing node attributes. As (SI-)VGRNN contribution is to have a model for diverse dynamic graph analytic tasks, the main goal of our method is node embedding. Hence, we are only interested in reconstructing the graph topology instead of the node attributes. This is a common practice in node embedding methods that use node attributes for better node embedding. Potential extensions with other decoders can be integrated with (SI-)VGRNN to construct the node attributes if needed. Regarding the dimension of variables (**R4**), as (SI-)VGRNN is a node embedding method for dynamic graphs, each node is embedded to a point in the latent space. Hence, the first dimension of $\mathbf{X}_t$ and $\mathbf{Z}_t$ are the same and the second dimension of $\mathbf{Z}_t$ is user specified latent dimension. If we reduce the first dimension of $\mathbf{Z}_t$, it would be "graph embedding" method rather than a "node embedding" technique, which is an interesting extension to our work.

Regarding the advantages of our work compared to related work (**R1**): 1) Dynamic network embedding is pursued with various techniques such as matrix factorization [Zhu et al.,2016], deep learning [Seo et al., 2016], and random walks [Yu et al., 2018], many of which are task specific methods and do not focus on representation learning. 2) Most existing methods either capture topological evolution or attribute changes to learn dynamic node embeddings [Yang et al., 2017;Sarkar et al., 2007] but only a few model both changes simultaneously [Trivedi et al., 2019]. 3) None of the existing methods model the uncertainty of the latent representations. While generative models in form of parametric temporal point processes [Trivedi et al., 2017] and deep temporal point processes [Trivedi et al., 2019] have been used for modeling dynamic graphs, to the best of our knowledge, (SI-)VGRNN is the first variational based deep generative model for representation learning of dynamic graphs. A more comprehensive related work section will be added.

[Meta-Review · NeurIPS 2019]

The proposed method is a sound combination of existing methods (GRNN [20] (deterministic) and GVAE [13] (no smoothness in time)), providing impressive performance gain for dynamic/new link prediction. Interpretability is stated but NOT supported by any. The authors should correct the paper for the camera ready so that all statements are supported.